# Sustained Elevated Blood Pressure Accelerates Atherosclerosis Development in a Preclinical Model of Disease

**DOI:** 10.3390/ijms22168448

**Published:** 2021-08-06

**Authors:** Andrés Gonzalez-Guerra, Marta Roche-Molina, Nieves García-Quintáns, Cristina Sánchez-Ramos, Daniel Martín-Pérez, Mariya Lytvyn, Javier de Nicolás-Hernández, José Rivera-Torres, Diego F. Arroyo, David Sanz-Rosa, Juan A. Bernal

**Affiliations:** 1Centro Nacional de Investigaciones Cardiovasculares (CNIC), 28029 Madrid, Spain; andres.gonzalez@embl.it (A.G.-G.); marta.roche@gmail.com (M.R.-M.); nieves.garcia@cnic.es (N.G.-Q.); cristina.sanchez@cnic.es (C.S.-R.); daniel.martin@cnic.es (D.M.-P.); mariya.lytvyn@cnic.es (M.L.); denicolasj@gmail.com (J.d.N.-H.); JOSE.RIVERA@universidadeuropea.es (J.R.-T.); diegof.arroyo.sspa@juntadeandalucia.es (D.F.A.); DAVID.SANZ@universidadeuropea.es (D.S.-R.); 2Facultad CC Biomédicas, Universidad Europea, 28670 Madrid, Spain; 3Servicio de Cardiología, Hospital Universitario Virgen Macarena, 41009 Sevilla, Spain; 4CIBERCV, 28029 Madrid, Spain

**Keywords:** renin, angiotensinogen, elevated blood-pressure, prehypertension, atherosclerosis, disease model, adeno associated virus (AAV), cardiovascular risk-factor

## Abstract

The continuous relationship between blood pressure (BP) and cardiovascular events makes the distinction between elevated BP and hypertension based on arbitrary cut-off values for BP. Even mild BP elevations manifesting as high-normal BP have been associated with cardiovascular risk. We hypothesize that persistent elevated BP increases atherosclerotic plaque development. To evaluate this causal link, we developed a new mouse model of elevated BP based on adeno-associated virus (AAV) gene transfer. We constructed AAV vectors to support transfer of the *hRenin* and *hAngiotensinogen* genes. A single injection of AAV-*Ren*/*Ang* (10^11^ total viral particles) induced sustained systolic BP increase (130 ± 20 mmHg, vs. 110 ± 15 mmHg in controls; *p* = 0.05). In *ApoE^−/−^* mice, AAV-induced mild BP elevation caused larger atherosclerotic lesions evaluated by histology (10-fold increase vs. normotensive controls). In this preclinical model, atheroma plaques development was attenuated by BP control with a calcium channel blocker, indicating that a small increase in BP within a physiological range has a substantial impact on plaque development in a preclinical model of atherosclerosis. These data support that non-optimal BP represents a risk for atherosclerosis development. Earlier intervention in elevated BP may prevent or delay morbidity and mortality associated with atherosclerosis.

## 1. Introduction

The global burden of disease attributable to hypertension is predicted to increase significantly from 26.4% measured in the year 2000 to above 29% by 2025 [1]. The impact of high blood pressure (BP) on the arterial tree includes thickening of artery walls, increased risk of rupture, and the development of atherosclerotic plaques. Despite this, to date, little attention has been paid to the interplay between BP and atherosclerosis development across a wide range of BP values (normal, high-normal, pre-hypertension, and overt hypertension). After years sharing a common definition in their guidelines, the American Heart Association/American College of Cardiology (AHA/ACC) [2] and the European Society of Cardiology (ESC) [3] have diverged in their approach since 2017, whereas the ESC maintained the previous definition (SBP 140–149 mmHg, DBP 90–99 mmHg) [3], the AHA/ACC adopted a lower threshold to define hypertension (SBP 130–139 mmHg, DBP 80–89 mmHg) [2]. These seemingly inconsistent criteria are partially resolved by the recommendation of both guidelines to control elevated BP by non-pharmacological interventions in the first instance and to start treating with antihypertensive drugs only in case the risk becomes high. Although the association between BP and cardiovascular events is linear [4,5], cut-offs are used to categorize BP as optimal, normal, high-normal, or hypertension [3]. Thus, while the question of how best to define optimal BP remains open, recent reports [6,7] highlight the importance of BP control, and the risk associated with high-normal BP. Whelton et al. reported that a rise in BP is positively associated with coronary artery calcium prevalence and the incidence of atherosclerosis-linked cardiovascular events using data from the Multi-Ethnic Study of Atherosclerosis (MESA) [7]. Among individuals with no traditional risk factors and with a systolic BP (SBP) <130 mmHg, the presence of atherosclerotic lesions and the risk of incident adverse events increase in-step with SBP increases above 90 mmHg [7]. This association between increasing BP categories and atherosclerosis has also been described in young low-risk individuals [6]. However, it is still controversial, which is the optimal BP goal and which patients may benefit from therapy [8,9,10].

Comprehensive understanding of the interaction between mild increase in BP and atherosclerosis requires experimental evidence based on new animal models of elevated BP [11,12] to test causal relationships. Nevertheless, to the best of our knowledge, no studies have developed chronic models of mild elevated BP to study plaque burden development. The aim of our study was to define the link between BP and atherosclerosis presence in multiple vascular territories, in particular, whether discrepancies in BP thresholds between AHA/ACC and ESC guidelines are differentially connected with atherosclerosis development. To study whether the link between a moderate BP increase and atherogenesis is mediated by a direct mechanical effect, we generated and analyzed a highly reproducible experimental mouse model based in adeno-associated virus (AAV) gene-transfer of mildly elevated BP in a proatherogenic background.

## 2. Results

### 2.1. Generation of a Model of Long-Term Elevated BP Using AAV-Ren/Ang Vectors

Recent reports provide insight into the association between BP and the prevalence of coronary artery calcium [6,7] and the incidence of cardiovascular events [7]. To determine whether the mechanical impact of mild BP elevation directly exacerbates atherosclerotic lesion progression, we generated a mouse disease model of chronic BP increase. An increase in BP is one of several pathophysiological processes linked to dysregulation of the renin-angiotensin-aldosterone system (RAAS) [13]. To modulate the RAAS, we generated two AAV vectors encoding human *Renin* (*Ren*) and *Angiotensinogen* (*Ang*) genes (Figure 1a). The AAV-*Ren* and AAV-*Ang* vectors were used to encapsidate viral particles in AAV serotype 9. A single intravenous injection of 10^11^ viral particles into wild-type C57BL/6J mice does not elicit any reported adverse responses in animals [14], and post-injection levels of serum alanine aminotransferase (ALT) and aspartate aminotransferase (AST) were similar in uninfected mice and mice infected with AAV9 viral particles (Appendix A). AAV viral infection and liver-specific ectopic expression of *Ren* and *Ang* thus did not induce hepatotoxicity. Hepatic co-transexpression of *Ren* and *Ang* significantly increased systolic BP in mice after 8 weeks. Quantitative RT-PCR analysis of liver samples after injection of AAV-*Ren* and AAV-*Ang* revealed a significant (*p* < 0.0001) increase in the *Ren* and *Ang* genes over background, confirming that the observed physiological effects were coupled to transcriptional activation (Figure 1b). Animals injected with only one of the AAV vectors remained normotensive compared with control animals transduced with empty vector (AAV-*Ren*, 107 ± 16 mmHg; AAV-*Ang*, 109 ± 15 mmHg; Control, 109 ± 15 mmHg). At 30 days post-injection, systolic BP in AAV- *Ren*/*Ang* transduced mice was significantly higher (130 ± 20 mmHg) than in mock-infected controls or mice expressing only one of the genes (109 ± 14 mmHg) (Figure 1c).

To obtain a comprehensive profile of arterial pressure, we recorded BP in mice over 28 weeks. In this long-term study, empty-vector-injected C57BL/6 mice maintained the baseline mean SBP of 110 ± 15 mmHg (Figure 1d), whereas in AAV-*Ren*/*Ang*-injected mice, the mean BP increased over the study period by 20 mmHg (to 130 ± 20 mmHg, *p* < 0.0001). Our results thus indicate that AAV-*Ren*/*Ang* co-transduction stimulates a sustained elevation in BP.

### 2.2. Elevation of BP Exacerbates Atherosclerosis in ApoE^−/−^ Mice Fed a High-Fat Diet (HFD)

To investigate the link between elevated BP and atherosclerosis development, we injected *ApoE*^−/−^ mice with AAV-*Ren* and AAV-*Ang* before switching the animals to a defined dietary regime. Thus, 8 weeks after infection, mice (BP, 133 ± 18 mmHg) were randomized to a HFD or standard chow for an additional 6 or 10 weeks (Figure 2a). As expected, 6 weeks after HFD initiation, serum total cholesterol (tChol) in AAV *Ren*/*Ang*-transduced HFD-fed *ApoE*^−/−^ mice was almost twice as high as in similar mice fed the standard chow diet (1202 ± 207 versus 607 ± 40 mg/dL) (Figure 2b). Serum low-density lipoprotein (LDL) showed the same pattern (372 ± 146 versus 76 ± 10 mg/dL) (Figure 2c). Notably, AAV *Ren*/*Ang* mice maintained elevated BP (137 ± 20 mmHg) at all analysed time points.

To assess the contribution of elevated BP to plaque burden in the context of hyperlipidaemia, we stained aortas from *ApoE*^−/−^ mice with Oil Red O. Lipid staining revealed lesions in the thoracic aortas, aortic arches, and secondary arterial branches of all HFD-fed AAV-*Ren*/*Ang*-transduced *ApoE*^−/−^ mice and to a greater extent than in the vessels of similarly fed empty vector transduced *ApoE*^−/−^ mice (Figure 2d,e). Histological analysis of the aortic sinus also revealed that lesions were more complex in fat-fed AAV-*Ren*/*Ang ApoE*^−/−^ mice, progressing well beyond fatty streaks (Figure 2f). Valve images of Trichrome stains demonstrated that most of the collagen is associated with the underlying internal area of the lesion, with collagen deposits more obvious within the core and in the arterial wall. With elevated BP, the volume of the atherosclerotic lesion increased, and vascular walls presented collagen deposition, which can cause stenosis in the plaque-associated vasculature. Plaque lipids are accumulated throughout the cap and into the core, with the greatest increase in the cap region where foam cells are visible using Trichrome and H&E staining. We also observed that elevation of BP provokes the development of necrotic core with multiple cholesterol clefts present in the plaque of aortic sinus from *ApoE*^−/−^ mice maintained on the HFD (Figure 2f).

Based on the morphometric evaluation of the H&E- and Trichome-stained section of aortic root plaques (Figure 3a), we confirmed significant differences in the total plaque area between HFD-fed *ApoE^−/−^* and AAV-*Ren**/Ang ApoE^−/−^* mice: 0.88 ± 0.08 (n = 9) vs. 0.45 ± 0.11 mm^2^ (n = 10), respectively (Figure 3b). The collagen content of aortic root plaques was also significantly different between groups (Figure 3c), although in advance plaque differences in the presence of areas of necrosis were not clearly noticeable (Figure 3d). Together, our results indicate that *Ren*/*Ang*-mediated elevation of BP by 20 mmHg to levels considered non-pathological in humans (120–140 mmHg) is a significant contributing factor to atherosclerosis development in an *ApoE*^−/−^ preclinical model.

### 2.3. The Mechanical Effect of a Mild elevation in BP Induces Atherosclerosis Independently of the Activation of the RAAS Pathway

To investigate whether atherosclerosis in AAV-*Ren*/*Ang* transduced mice is driven by the mechanical effect of BP elevation to 120–140 mmHg, we explored the effect of blocking the BP elevation without affecting RAAS pathway activity. For these experiments, we used the anti-hypertensive compound amlodipine, a long-acting calcium channel blocker unrelated to RAAS [15]. *ApoE*^−/−^ mice were injected with AAV-*Ren*/*Ang* and maintained on amlodipine (2.5 mg/kg) or saline for 10 weeks. Amlodipine treatment prevented the sustained BP increase in AAV-*Ren*/*Ang*-injected *ApoE*^−/−^ mice on HFD (Figure 4a). Correlated with the normal BP readings, aortic tissue from all saline-treated HFD-fed AAV-*Ren*/*Ang* -transduced mice contained lesions along the entire thoracic aorta, aortic arch, and secondary arterial branches, whereas vessels from amlodipine-treated AAV-*Ren*/*Ang* mice were lesion free (Figure 4b,c). Furthermore, analysis of plaque cross-sectional area at the level of the aortic arch showed that only saline-treated AAV-*Ren*/*Ang* mice developed massive plaques characterized by fat accumulation (0.335 ± 0.098 mm^2^). In contrast, the plaque cross-sectional area in amlodipine-treated AAV-*Ren*/*Ang ApoE*^−/−^ mice was negligible, showing no significant difference from controls (0.029 ± 0.019 mm^2^) (Figure 4d,e). These results demonstrate that a small increase in BP within a physiological range has a substantial impact on plaque development in a preclinical model of atherosclerosis.

## 3. Discussion

Two complementary findings summarize our research. Firstly, using a preclinical model of atherosclerosis, we show that the mechanical effect of mild BP elevation directly promotes atherosclerotic lesion progression independent of the activation of the RAAS pathway. Secondly, the causal relationship between the mechanical effect of mild BP elevation below the threshold for hypertension [3] and atherosclerotic plaque development highlights the importance of BP control. The AAV-*Ren**/Ang ApoE^−/−^* model thus presents a useful tool for the study of genetic and pharmacological interventions to control BP changes and define their consequences for disease development. Although our results cannot be directly extrapolated to human patients, we believe they serve as a solid basis for the development of well-designed clinical trials with clearly specified efficacy criteria. However, our findings endorse the value of BP monitoring and primordial prevention. 

Before the present study, a mouse model of sustained elevated BP in the “non-pathological range” has been lacking. A major advantage of AAV-mediated transexpression is its robust stability after a single administration [14]. Compared with the classical model of RAAS-driven hypertension based on Angiotensin-II (Ang-II) infusion with minipumps [16], the AAV-*Ren**/Ang* method yielded consistently lower (20 mmHg vs. 40–60 mmHg) [17,18] and more sustained BP elevations, without the need for invasive surgical procedures and their possible complications. Osmotic minipumps allow continuous infusion of Ang-II at a predetermined dose in diverse animal models; however, this approach is time limited to a few weeks, preventing its application in chronic studies on the long-term effects of BP dysregulation over several months. In contrast, expression of AAV-*Ren**/Ang* genes driven by a liver-specific promoter could be detected 6 months after transduction in all injected animals, with no evidence of hepatotoxicity or activation of the inflammatory response.

Our results suggest that AAV-*Ren**/Ang*-transduced mice could be a useful platform for testing specific antihypertensive therapies and demonstrate that AAV-transfer methodology has the potential to make valuable contributions to the specific understanding of the relationship between BP and atherosclerosis. The clear effect of elevated BP on atheroma plaque development in the AAV-*Ren**/Ang* model could easily be used to test genetic interactions in combination with specific genetic models without the need for tedious, costly, and time-consuming backcrosses, as we have demonstrated here with *ApoE^−/−^* mice.

To our knowledge, no previous studies have demonstrated the ability of chronic BP elevation (20 mmHg) alone to accelerate atherosclerosis. In our analysis, atherosclerosis development in HFD-fed AAV-*Ren**/Ang*-injected *ApoE^−/−^* mice on the C57BL/6J genetic background was accelerated compared to *ApoE^−/−^* mice injected with empty vector; however, AAV-*Ren**/Ang ApoE^−/−^* and *ApoE^−/−^* mice on the atherogenic diet showed no differences in serum cholesterol and triglyceride or body weight, indicating that these parameters did not influence the observed difference in atherosclerosis development. The AAV approach used here can be used to develop progressive and chronic models of other diseases complicated by elevated BP and on other genetic backgrounds, enabling the analysis of genetic interactions and potential antihypertensive therapies.

Our findings thus open a new field for devising personalized lifestyle recommendations for asymptomatic individuals at risk of developing subclinical atherosclerosis. Further research is required to investigate whether elevated BP should be considered a high cardiovascular risk condition and to determine the specific BP treatment threshold for the asymptomatic population.

### 3.1. Study limitations

Investigating basic biology and interventions to modify the development of atherosclerosis in mouse models often prompts discussions about the validity and translational value of the findings to humans. In the present study, we demonstrate a causal link between BP elevation and atheroma plaque development. BP elevation is a physical parameter, and most likely a common trigger in mouse and human vascular pathophysiology, suggesting that the results of this study are more likely to translate to humans. Although our mouse model of AAV-*Ren**/Ang* transduction is an effective approach to elevate BP in the physiological range (~20 mmHg) for prolonged periods (>6 months), one study limitation is that the model is based on the artificial expression of human *Renin* and *Angiotensinogen* in the liver. In its classical view, liver-produced angiotensinogen is cleaved in the plasma by the tightly regulated renin produced by the kidney. Our model therefore cannot be used to study endogenous regulation of the RAAS pathway. 

### 3.2. Conclusions

Our results demonstrate that a mild elevation in BP is directly linked to atherosclerosis development in a preclinical model of atherogenesis. These data support the idea that non-optimal BP is a risk factor for the progress of atherosclerosis. Early intervention to reduce BP to the optimal range might be of interest for the prevention of atherosclerosis and subsequent CV events. Further research is needed to determine whether anti-hypertensive medication and/or lifestyle interventions targeting dietary habits, physical activity, and body weight might have a beneficial impact on atherosclerosis development in individuals with non-optimal BP.

## 4. Materials and Methods

### 4.1. Animal Experimental Design 

All animal procedures conformed to the guidelines from Directive 2010/63/EU of the European Parliament on the protection of animals used for scientific purposes. Animal experiments were carried out in accordance with the CNIC Institutional Ethics Committee recommendations and were approved by the Animal Experimentation Committee (Scientific Procedures) of Comunidad de Madrid (project number PROEX 019/17). Wild-type adult male mice and homozygous *ApoE*-deficient mice (*ApoE^−/−^*), both on the C57BL/6J genetic background, were originally obtained from Jackson Laboratories (Charles River Laboratories). *ApoE^−/−^* mice were used because of their susceptibility to atheroma plaque development. A single injection of 10^11^ viral particles of AAV-*Ren* and AAV-*Ang* viruses (AAV-*Ren**/Ang*) induced a sustained increase in systolic BP (SBP) (SBP 130 ± 20 mmHg, vs. 110 ± 15 mmHg in control mice; *p* = 0.05). Control mice were injected with an AAV-empty vector. For atherosclerosis assessment, mice were switched from a standard low-fat rodent diet (reference 2014, Teklad global rat/mouse chow, Harlan Interfauna) to a high-fat diet (HFD) containing 0.75% cholesterol (reference S8492-E010, Ssniff). The end-point analysis was 10 weeks after the diet switch, when the aortic root and descending aorta were isolated and analyzed for atheroma plaque formation. Animals were randomly assigned to experimental groups (simple randomization in 2020 using the Research Randomizer web page https://www.randomizer.org (accessed on 15 June 2021)). Mice were individually housed in wire-bottomed cages in a temperature-controlled room (22 ± 0.8 °C) with a 12 h light–dark cycle and a relative humidity of 55 ± 10%. Mice had free access to food and water. Atherosclerotic plaques were measured by experimenters blinded to the experimental group. Blood pressure was reduced by treating mice with amlodipine (2.5 mg/kg) in drinking water (12.5 mg/L) for the indicated time, with the drug being refreshed once a week. All animals were included in the analysis.

### 4.2. Adeno-Associated Virus (AAV) Vector Production and Purification

AAV vectors were produced by the triple transfection method using HEK293T cells, as described previously [19,20]. The shuttle plasmid pAAV-*Ren**/Ang*, derived from *pAcTnT*, was packaged into AAV-9 capsids using the helper plasmids pAdDF6 (providing the three adenoviral helper genes) and plasmid pAAV2/9 (providing the rep and cap viral genes) (PennVector). The pAAV-*Ren**/Ang* fragment, containing the human *Ren**in* and *Ang**iotensinogen* genes (*Ren**/Ang*) under the liver-specific promoter HCR-hAAH2, was co-transfected with the helper plasmids into HEK293T cells using linear polyethylenimine (*M*_W_ 25,000). The cells were seeded in Hyperflasks (Corning) at 1.2 × 10^8^ cells per flask the day before transfection. A total of 840 μg of an equimolar mix of plasmid DNA was added to each Hyperflask. At 72 h after transfection, the cells were collected by centrifugation, and the cell pellet was resuspended in TMS (50 mmol/L Tris HCl, 150 mmol/L NaCl, 2 mmol/L MgCl_2_) on ice before digestion with benzonase nuclease (150 units/mL; Millipore) at 37 °C for 30 min. Viral particles contained in the clarified supernatants were purified by iodixanol gradient centrifugation [21]. Gradient fractions containing virus were concentrated using Amicon UltraCel columns (Millipore) and stored at −70 °C.

### 4.3. Determination of AAV Vector Titter 

AAV vector titters (viral genomes per mL) were determined by quantitative real-time PCR [14] using the following primers: *Ren**in* Fw 5′GGAACAGAACTCACCCTCCG 3′; *Ren**in* Rv, 5′ GTGATTCCACCCACGGTGAT 3′; *Ang**iotensinogen* Fw, 5′ AACTGGTGCTGCAAGGATCT 3′; *Ang**iotensinogen* Rv 5′ TTCAGCTCGGTGTGCAGAAT 3′. Standard curves were constructed from known copy numbers (10^5^–10^8^) of the plasmids (*pAAV-HRC-hAAT-Ren/Ang*) carrying the appropriate complementary DNA.

### 4.4. Quantitative Real-Time PCR 

For mouse tissue analysis, total RNA was isolated from mouse livers using the Directzol RNA Miniprep Kit (Zymo, Irvine, CA, USA) and reverse transcribed with the High-Capacity cDNA Reverse Transcription Kit (Applied Biosystems). The complementary DNAs were then used for real time PCR using the Power SYBR Green PCR Master Mix (Applied Biosystems, Waltham, MA, USA). Amplification, detection, and data analysis were performed with an ABI PRISM 7900HT Sequence Detection System. The crossing threshold values for individual samples were normalized to *Gapdh*. Changes in mRNA content are expressed as the fold change relative to the control. We used the following control primers: *Gapdh* Fw, 5′-TTGATGGCAACAATCTCCAC-3′; *Gapdh* Rv, 5′-CGTCCCGTAGACAAAATGGT-3′.

### 4.5. Serum Analysis 

Overnight-fasted serum total cholesterol (tChol), low-density lipoprotein (LDL), alanine aminotransferase (ALT), aspartate transaminases (AST), lactate dehydrogenase (LDH), and creatine kinase (CK) were assayed with a Dimension R × L Max HM clinical chemistry system from SIEMENS. Lipid amounts are expressed as milligrams per decilitre (mg/dL) and enzyme activities as units per litter (U/L).

### 4.6. Blood Pressure (BP) Measurements in Mice 

Arterial BP was measured by the mouse-tail cuff method using the automated BP-2000 Blood Pressure Analysis System (Visitech Systems, Apex, NC, USA). Mice were trained for BP measurements every day for one week. After training, the measurements were repeated several times during the experiments, with measurements taken on at least three consecutive days at the same time of day. For BP measurements, mice were located in a tail cuff limiter on a heated surface (37 °C). Fifteen consecutive SBP measurements were recorded, and the last 10 readings for each mouse were used for analysis.

### 4.7. Analysis of Atherosclerotic Lesions in ApoE-Deficient Mice 

To quantify the surface area occupied by atherosclerotic plaques, aortas were stained with Oil Red O and Masson’s trichrome and then analyzed by quantitative morphometry. Mouse hearts and aortas were perfused with PBS, removed, fixed in 4% paraformaldehyde for 24 h, incubated 24 h in PBS supplemented with 30% sucrose, and embedded in OCT and cryopreserved at −70 °C. Cryocut cross-sections (5 μm) were then prepared.

### 4.8. Statistics 

Experiments were designed to use the minimum number of mice needed to give sufficient statistical power, and the numbers of animals used are comparable to published literature for similar assays. No animals were excluded from the analyses. Animal data were analyzed by one-way ANOVA, two-way ANOVA, and Student’s *t*-test. Error bars represent SEM. In all corresponding figures, * *p* ≤ 0.05, ** *p* ≤ 0.01, *** *p* ≤ 0.001, **** *p* ≤ 0.0001, and ns *p* > 0.05.

## Figures and Tables

**Figure 1 ijms-22-08448-f001:**
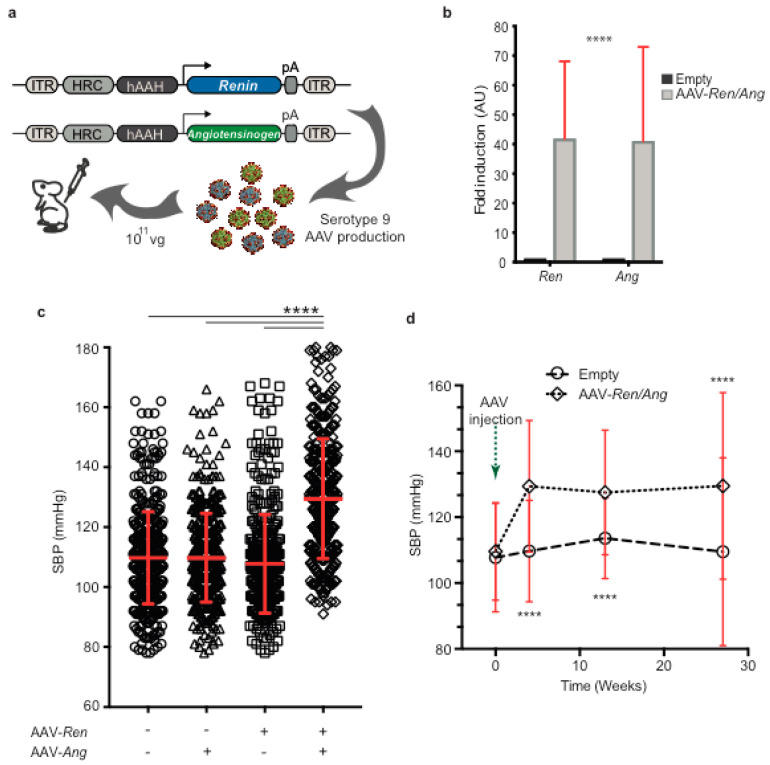
A single injection of AAV-*Ren**/Ang* viruses increases systolic blood pressure (SBP). (**a**) Plasmids encoding the expression of *hRenin* (*Ren*) and *hAngiotensinogen* (*Ang*). The plasmids were introduced into serotype 9 adeno-associated viruses (AAV) and injected via the tail vein. (**b**) Liver mRNA expression of AAV-delivered exogenous *Ren* and *Ang* genes in C57BL6/J mice. (**c**) C57BL6/J mice were given either a single injection of AAV-*Ren* or AAV-*Ang* or a combined injection with both viruses (5 × 10^10^ vector genomes/animal, n = 10). Co-injection induced a sustained maximal mean SBP increase of ≈20 mmHg. Mean SBP was calculated from measurements from weeks 8 to 12. Data are mean ± SEM. **** *p* < 0.0001 versus empty vector transduced control group. (**d**) Evolution of tail-cuff measured SBP after AAV-*Ren**/Ang* or AAV-empty vector injection. **** *p* < 0.0001, 2-way repeated-measures ANOVA (n = 10–25 per group). ITR, inverted terminal repeats; HCR-hAAH, human antitrypsin gene promoter with an *ApoE* gene enhancer; pA, poly-A sequence.

**Figure 2 ijms-22-08448-f002:**
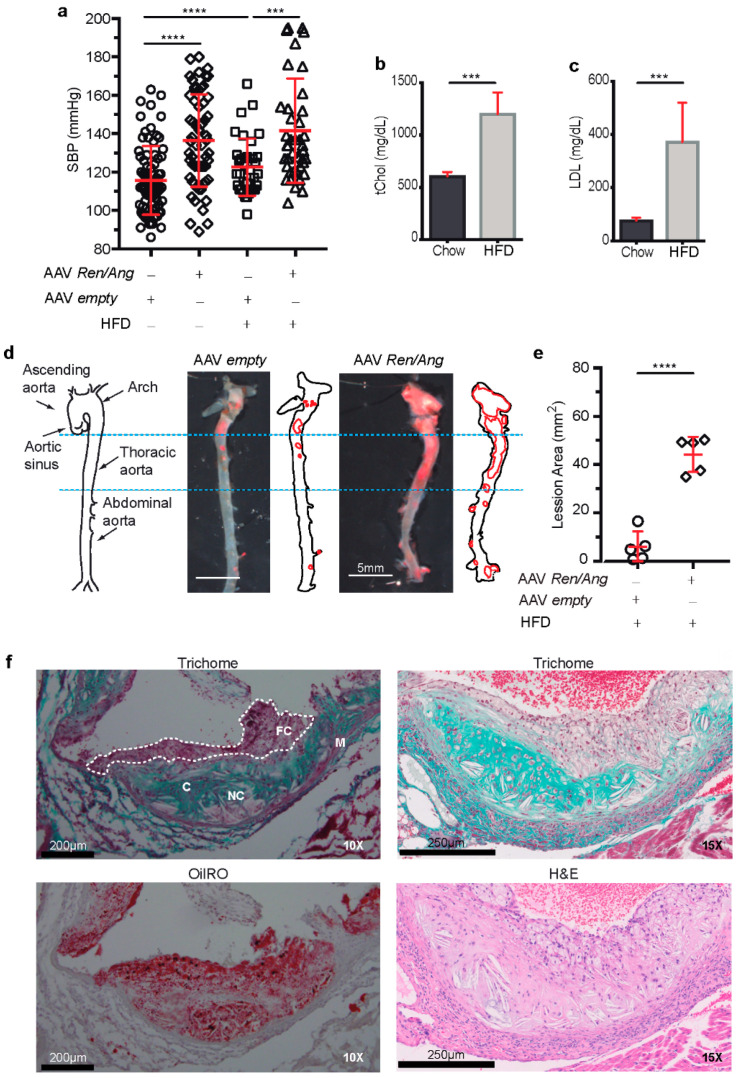
Elevated blood pressure (BP) induces atheroma plaque development in *ApoE^−/−^* mice. (**a**) SBP in *ApoE^−/−^* mice 6 weeks after injection with AAV-*Ren**/Ang* or AAV-empty vector and after subsequent maintenance for a further 10 weeks on a high fat diet (HFD) (n = 15 animals per group). Data are mean ± SEM. *** *p* < 0.001 and **** *p* < 0.0001, 2-way ANOVA followed by Bonferroni post-test. (**b**) Serum total cholesterol and (**c**) serum low-density lipoprotein (LDL) cholesterol in AAV-*Ren**/Ang*-injected *ApoE^−/−^* mice fed standard chow or the HFD for 10 weeks starting 6 weeks after AAV injection. (**d**) Schematic representation of the aorta. Aortic sinus, ascending aorta, arch, thoracic aorta, and abdominal aorta are indicated with black arrows. Representative complete aortas from AAV-empty control and AAV-*Ren**/Ang*-injected *ApoE^−/−^* mice stained with Oil Red O. Aortas were extracted 10 weeks after starting the HFD. Scale bar, 5 mm. (**e**) Lesion area in aortas stained as in D. Data in (**b**,**c**,**e**) are mean ± SEM. *** *p* < 0.001, **** *p* < 0.0001, Student *t*-test. (**f**) Representative Masson’s trichrome, Oil Red O, and H&E staining of the aortic sinus in HFD-fed AAV-*Ren**/Ang ApoE^−/−^* mice. FC, foam cells; NC, necrotic core; C, collagen lesions; M, tunica media. Scale bars, 200 μm (10×) and 250 μm (15×).

**Figure 3 ijms-22-08448-f003:**
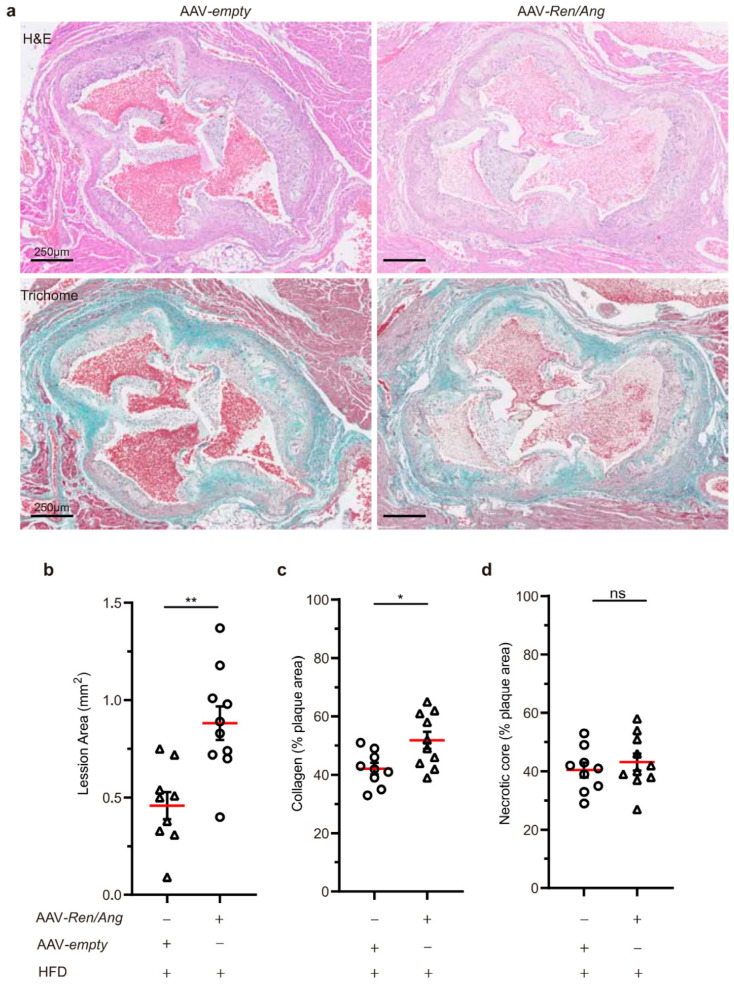
Non-optimal BP increases atherosclerosis at the aortic root. (**a**) Representative whole view of aortic roots from AAV-empty control and AAV-*Ren**/Ang*-injected *ApoE^−/−^* mice stained with H&E and Masson’s trichrome. Aortas were extracted 10 weeks after starting the HFD. Scale bar, 250 μm. (**b**) Quantification of atherosclerotic plaque area from a (n = 9–10 AAV-empty and AAV-*Ren**/Ang* mice) indicated as mm^2^. (**c**,**d**) Quantification of collagen and necrotic core area expressed as a percentage of the total plaque cross-sectional area (n = 9-10). Data in (**b**–**d**) are mean ± SEM. * *p* < 0.05, ** *p* < 0.01, ns, non-significant, Student *t*-test.

**Figure 4 ijms-22-08448-f004:**
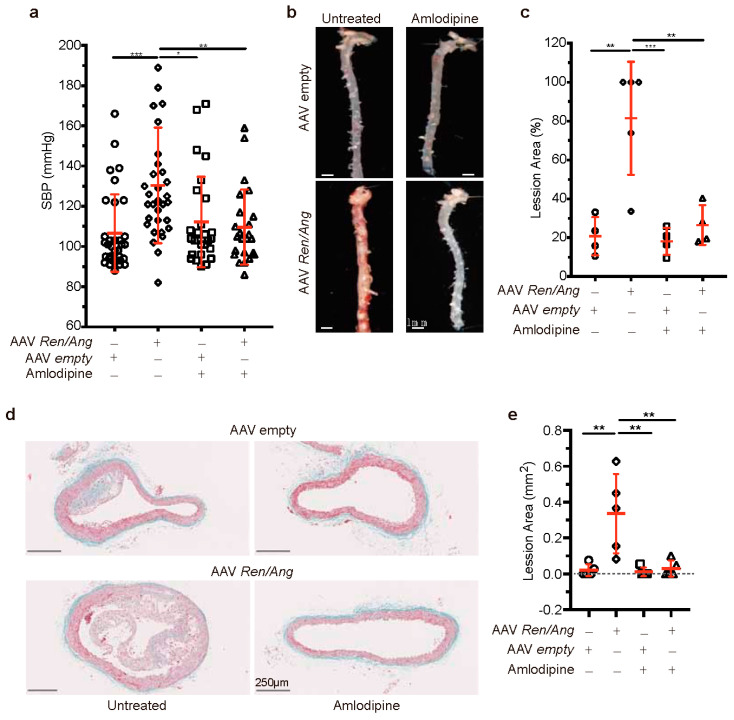
The calcium antagonist amlodipine reduces blood pressure (BP) and atherosclerosis induced by AAV-*Ren**/Ang* transduction in mice. (**a**) SBP (mmHg) in *ApoE^−/−^* mice injected with AAV-empty shuttle vector or AAV-*Ren**/Ang*. Mice were maintained for 10 weeks on the high-fat diet (HFD) starting 6 weeks after AAV injection and were untreated or treated with amlodipine (2.5 mg/kg) in the drinking water. SBP was measured at the same time of the day in all groups. Data are mean ± SEM. * *p* < 0.05, ** *p* < 0.01, *** *p* < 0.001, 2-way ANOVA followed by Bonferroni post-test (n = 10 per group). (**b**) Representative Oil Red O-stained aortas from AAV-empty shuttle vector and AAV-*Ren**/Ang*-injected *ApoE^−/−^* mice untreated or treated with amlodipine throughout the 12-week HFD period. Scale bar, 1 mm. (**c**) Lesion area in aortas stained as in (**b**). Data are mean ± SEM. ** *p* < 0.01, *** *p* < 0.001, one-way ANOVA with Tukey’s multiple comparison test (n = 4-5 mice per group). Each data point represents an individual aorta, horizontal red bars denote mean values, and black bars denote SEM. (**d**) Representative Masson’s trichrome staining of descending aorta sections from *ApoE^−/−^* mice injected with empty vector or AAV-*Ren**/Ang* and untreated or treated with amlodipine. (**e**) Lesion area in sections stained as in d. Data are mean ± SEM. ** *p* < 0.01, one-way ANOVA with Tukey’s multiple comparison test (n = 4–5 mice per group). Each data point represents an individual aorta, horizontal red bars denote mean values, and bars denote SEM.

## Data Availability

All the data supporting this work are available within the article and its Appendix A, or can be obtained from the corresponding author upon reasonable request.

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
