# Peer review of "Sustained Elevated Blood Pressure Accelerates Atherosclerosis Development in a Preclinical Model of Disease"

_ijms, 2021, doi:10.3390/ijms22168448_

Round 1

Reviewer 1 Report

Summary:

The authors have generated a novel mildly elevated blood pressure mouse model using AAV9 expression of renin and angiotensin II in the liver.  The blood pressure is around 20 mmHg higher than the normal control and yet caused severe atherosclerosis in the apoE -/- mice fed with a high-fat diet. Lowering the BP with amlodipine prevented the development of atherosclerosis in these mice.

Major points:

This is a very well-designed study and provided evidence that high normal blood pressure might be a therapeutic target to lower the incidence of coronary artery disease. Although the use of renin and angiotensin II expression in the liver is a limitation (acknowledged by the authors), the evidence of high normal blood pressure causes atherosclerosis in the mouse model is compelling. This provided evidence for targeting a lower blood pressure to prevent atherosclerosis.

Minor points:

  1. The introduction could be expanded with the information of the first part of the discussion (e.g. controversies of the targeted BP by ESC and AHA ) to give a better rationale to do the experiments and catch the attention of the readers.
  2. Fig 1. Are these mean±SEM or mean±STD? It looks like STD given the wide variations with n=10-25 per group.
  3. Line 214. It should be “….atherosclerosis independent of the activation of the RAAS pathway.”

Author Response

We are pleased that the reviewers have pointed out the high quality and the importance of our work. We thank them for their constructive comments, which have helped us to strengthen the manuscript.

In revising the same, we provide new data and a point-by-point response addressing all the Reviewers’ comments.

Reviewer 1

The authors have generated a novel mildly elevated blood pressure mouse model using AAV9 expression of renin and angiotensin II in the liver.  The blood pressure is around 20 mmHg higher than the normal control and yet caused severe atherosclerosis in the apoE -/- mice fed with a high-fat diet. Lowering the BP with amlodipine prevented the development of atherosclerosis in these mice.

We thank the reviewer for his/her comments that have helped us to improve our manuscript.

Major points:

This is a very well-designed study and provided evidence that high normal blood pressure might be a therapeutic target to lower the incidence of coronary artery disease. Although the use of renin and angiotensin II expression in the liver is a limitation (acknowledged by the authors), the evidence of high normal blood pressure causes atherosclerosis in the mouse model is compelling. This provided evidence for targeting a lower blood pressure to prevent atherosclerosis.

Minor points:

The introduction could be expanded with the information of the first part of the discussion (e.g. controversies of the targeted BP by ESC and AHA ) to give a better rationale to do the experiments and catch the attention of the readers.

We agree with the Reviewer that the introduction would benefit from being expanded. We have added the requested paragraphs as follows:

“INTRODUCTION:

The global burden of disease attributable to hypertension is predicted to increase significantly from 26.4% measured in year 2000 to above 29% by 2025[1].  The impact of high blood pressure (BP) on the arterial tree includes thickening of artery walls, increased risk of rupture, and the development of atherosclerotic plaques. Despite this, to date little attention has been paid to the interplay between BP and atherosclerosis development across a wide range of BP values (normal, high-normal, pre-hypertension, and overt hypertension). After years sharing a common definition in their guidelines, the American Heart Association/American College of Cardiology (AHA/ACC)[2] and the European Society of Cardiology (ESC)[3] have diverged in their approach since 2017.  Whereas the ESC maintained the previous definition (SBP 140-149 mmHg, DBP 90–99 mmHg)[3], the AHA/ACC adopted a lower threshold to define hypertension (SBP 130-139 mmHg, DBP 80-89 mmHg)[2]. These seemingly inconsistent criteria are partially resolved by the recommendation of both guidelines to control elevated BP by non-pharmacological interventions in the first instance and to start treating with antihypertensive drugs only in case the risk becomes high. Although the association between BP and cardiovascular events is linear[4, 5], cut-offs are used to categorize BP as optimal, normal, high-normal, or hypertension[3]. Thus, while the question of how best to define optimal BP remains open, recent reports[6, 7] highlight the importance of BP control, and the risk associated with high-normal BP. Whelton et al reported that a rise in BP is positively associated with coronary artery calcium prevalence and the incidence of atherosclerosis-linked cardiovascular events using data from the Multi-Ethnic Study of Atherosclerosis (MESA)[7]. Among individuals with no traditional risk factors and with a systolic BP (SBP) <130 mmHg, the presence of atherosclerotic lesions and the risk of incident adverse events increase in-step with SBP increases above 90mmHg[7]. This association between increasing BP categories and atherosclerosis has also been described in young low-risk individuals[6]. However, it is still controversial which is the optimal BP goal and which patients may benefit from therapy[8-10].

Comprehensive understanding of the interaction between mild increase in BP and atherosclerosis requires experimental evidences based on new animal models of elevated BP [11, 12] to test causal relationships. Nevertheless, to the best of our knowledge, no studies have developed chronic models of elevated BP to study plaque burden development. The aim of our study was to define the link between BP and atherosclerosis presence in multiple vascular territories, in particular, whether discrepancies in BP thresholds between AHA/ACC and ESC guidelines are differentially connected with atherosclerosis development. To study whether the link between a moderate BP increase and atherogenesis is mediated by a direct mechanical effect, we generated and analysed a highly reproducible experimental mouse model based in adeno-associated virus (AAV) gene-transfer of mildly elevated BP in a proatherogenic background.”

Fig 1. Are these mean±SEM or mean±STD? It looks like STD given the wide variations with n=10-25 per group.

We have used mean±SEM in our Graph representation of the Figure 1 data.

Line 214. It should be “….atherosclerosis independent of the activation of the RAAS pathway.”

We have corrected the text as requested.

Reviewer 2 Report

The manuscript presents evidence demonstrating that a moderate increase in blood pressure can drive atherogenesis in a preclinical model of atherosclerosis.

This is a valuable study on a clinically relevant topic and well written, however some minor queries need to be considered:

  • The introduction is well written, but in this section the aim of the study needs to be properly highlighted and justified (significance/implications of your findings/conclusions). The introduction is too short and does not provide sufficient information regarding the scope of the research, its originality, noveltyand priority, the context for your paper and what is already present in the literature. The introduction should provide sufficient background information for the reader to be able to follow the information presented and inform the reader about how that information will be presented. It is usually half to three-quarters of a page in length.

  • Results: Histopathological analysis: please, explain in depeer detail and appropriately the structural/morphological features you are talking about. Figure 2 (LM images) Please replace by a higher resolved image – foam cells are is or hardly recognizable. Please specify the significance of collagen lesions. Authors should provide a clear description of the structural features of aortas.

  • The are typographical and grammatical errors throughout the text that must be corrected (i.e. Abstract line 27: “associated to atherosclerosis” should be corrected as follows: “associated with atherosclerosis”)

Author Response

We are pleased that the reviewers have pointed out the high quality and the importance of our work. We thank them for their constructive comments, which have helped us to strengthen the manuscript.

In revising the same, we provide new data and a point-by-point response addressing all the Reviewers’ comments.

Reviewer 2

The manuscript presents evidence demonstrating that a moderate increase in blood pressure can drive atherogenesis in a preclinical model of atherosclerosis.

We thank the reviewer for his/her suggestions that have helped us to strengthen our work.

This is a valuable study on a clinically relevant topic and well written, however some minor queries need to be considered:

  • The introduction is well written, but in this section the aim of the study needs to be properly highlighted and justified (significance/implications of your findings/conclusions). The introduction is too short and does not provide sufficient information regarding the scope of the research, its originality, novelty and priority, the context for your paper and what is already present in the literature. The introduction should provide sufficient background information for the reader to be able to follow the information presented and inform the reader about how that information will be presented. It is usually half to three-quarters of a page in length.

We have extended the introductory section as requested. We believe that the Introduction now provides sufficient background to understand the significance and the originality of our study, as well as the importance of our conclusions.

INTRODUCTION:

The global burden of disease attributable to hypertension is predicted to increase significantly from 26.4% measured in year 2000 to above 29% by 2025[1].  The impact of high blood pressure (BP) on the arterial tree includes thickening of artery walls, increased risk of rupture, and the development of atherosclerotic plaques. Despite this, to date little attention has been paid to the interplay between BP and atherosclerosis development across a wide range of BP values (normal, high-normal, pre-hypertension, and overt hypertension). After years sharing a common definition in their guidelines, the American Heart Association/American College of Cardiology (AHA/ACC)[2] and the European Society of Cardiology (ESC)[3] have diverged in their approach since 2017.  Whereas the ESC maintained the previous definition (SBP 140-149 mmHg, DBP 90–99 mmHg)[3], the AHA/ACC adopted a lower threshold to define hypertension (SBP 130-139 mmHg, DBP 80-89 mmHg)[2]. These seemingly inconsistent criteria are partially resolved by the recommendation of both guidelines to control elevated BP by non-pharmacological interventions in the first instance and to start treating with antihypertensive drugs only in case the risk becomes high. Although the association between BP and cardiovascular events is linear[4, 5], cut-offs are used to categorize BP as optimal, normal, high-normal, or hypertension[3]. Thus, while the question of how best to define optimal BP remains open, recent reports[6, 7] highlight the importance of BP control, and the risk associated with high-normal BP. Whelton et al reported that a rise in BP is positively associated with coronary artery calcium prevalence and the incidence of atherosclerosis-linked cardiovascular events using data from the Multi-Ethnic Study of Atherosclerosis (MESA)[7]. Among individuals with no traditional risk factors and with a systolic BP (SBP) <130 mmHg, the presence of atherosclerotic lesions and the risk of incident adverse events increase in-step with SBP increases above 90mmHg[7]. This association between increasing BP categories and atherosclerosis has also been described in young low-risk individuals[6]. However, it is still controversial which is the optimal BP goal and which patients may benefit from therapy[8-10].

Comprehensive understanding of the interaction between mild increase in BP and atherosclerosis requires experimental evidences based on new animal models of elevated BP [11, 12] to test causal relationships. Nevertheless, to the best of our knowledge, no studies have developed chronic models of elevated BP to study plaque burden development. The aim of our study was to define the link between BP and atherosclerosis presence in multiple vascular territories, in particular, whether discrepancies in BP thresholds between AHA/ACC and ESC guidelines are differentially connected with atherosclerosis development. To study whether the link between a moderate BP increase and atherogenesis is mediated by a direct mechanical effect, we generated and analysed a highly reproducible experimental mouse model based in adeno-associated virus (AAV) gene-transfer of mildly elevated BP in a proatherogenic background.”

  • Results: Histopathological analysis: please, explain in depeer detail and appropriately the structural/morphological features you are talking about. Figure 2 (LM images) Please replace by a higher resolved image – foam cells are is or hardly recognizable. Please specify the significance of collagen lesions. Authors should provide a clear description of the structural features of aortas.

Following the reviewer’s request for clarification we have added new text and data on the histopathological description of the atherosclerotic lesion (Figure 2F).

…” Valve images of Trichrome stains demonstrated that most of the collagen is associated with the underlying internal area of the lesion, with collagen deposits more obvious within the core and in the arterial wall. With elevated BP, the volume of the atherosclerotic lesion increased, and vascular walls presented collagen deposition what can cause stenosis in the plaque associated vasculature. Plaque lipids are accumulated throughout the cap and into the core, with the greatest increase in the cap region where foam cells are visible. We also observed that elevation of BP provokes the development of necrotic core with multiple cholesterol clefts present in the plaque of aortic sinus from ApoE-/- mice maintained on the HFD (Figure 2F).”…

  • The are typographical and grammatical errors throughout the text that must be corrected (i.e. Abstract line 27: “associated to atherosclerosis” should be corrected as follows: “associated with atherosclerosis”)

The text has now been professionally edited.
